# [F18]FDG PET/CT-Derived Metabolic and Volumetric Biomarkers Can Predict Response to Treatment in Locally Advanced Cervical Cancer Patients

**DOI:** 10.3390/cancers14184382

**Published:** 2022-09-08

**Authors:** Ronit Gill, Roxolyana Abdah-Bortnyak, Amnon Amit, Uval Bar-Peled, Zohar Keidar

**Affiliations:** 1Nuclear Medicine Department, Rambam Health Care Campus, Haifa 3109601, Israel; 2Oncology Department, Rambam Health Care Campus, Haifa 3109601, Israel; 3Bruce Rappaport Faculty of Medicine, Technion—Israel Institute of Technology, Haifa 3200003, Israel; 4Obstetrics and Gynecology Department, Rambam Health Care Campus, Haifa 3109601, Israel

**Keywords:** cervical cancer, FDG, PET/CT, TLG, MTV

## Abstract

**Simple Summary:**

Carcinoma of the uterine cervix is one of the most common and severe malignancies in women. Up to 40% of locally advanced cervical cancer patients treated with definitive chemoradiation therapy will not respond or will develop disease recurrence. The aim of this retrospective study was to evaluate the relationship between quantitative metabolic and volumetric parameters extracted from FDG PET/CT imaging data and the response rate to definitive chemoradiation therapy in this group of patients. FDG PET/CT studies of 90 cervical cancer patients were analyzed, and it was found that quantitative metabolic and volumetric parameters such as SUVmax, SUVmean, MTV, and TLG are higher in locally advanced cervical cancer patients who will not respond to definitive chemoradiation therapy. Specifically, in patients who are not metastatic at staging, MTV and TLG values can serve as a predictor for treatment response and thus may alter treatment strategy.

**Abstract:**

(1) Purpose: Current study aimed at evaluating the relationship between quantitative metabolic and volumetric FDG PET/CT parameters and the response to definitive chemoradiation therapy in locally advanced cervical cancer patients; (2) Methods: Ninety newly diagnosed locally advanced cervical cancer patients (FIGO IB2-IVA) were investigated. All patients underwent PET/CT at staging and after treatment. Metabolic and volumetric parameters, including SUVmax, SUVmean, Total Lesion Glycolysis (TLG), and Metabolic Tumor Volume (MTV) of the primary tumor and metastatic lymph nodes were measured and compared between patients with and without complete metabolic response (CMR). A similar analysis was performed in a subgroup of FIGO IB2-IIB patients; (3) Results: SUVmax and SUVmean of the primary tumor as well as those of metastatic lymph nodes, MTV, and TLG were found to be significantly different between CMR and non-CMR patients. In a subgroup of patients with FIGO IB2-IIB disease, MTV and TLG identified women who will achieve CMR with a threshold of 31.1 cm^3^ for MTV and 217.8 for TLG; (4) Conclusions: PET/CT-derived quantitative metabolic and volumetric parameters are higher in locally advanced cervical cancer patients who will not respond to definitive chemoradiation therapy. Specifically, in patients who are not metastatic at staging, MTV and TLG values can serve as a predictor for treatment response and thus may alter treatment strategy.

## 1. Introduction

Carcinoma of the uterine cervix is one of the most common and severe malignancies in women [1]. Treatment of cervical cancer is decided according to the stage of the disease. While definitive chemoradiation is a largely accepted treatment for locally advanced cervical cancer patients (FIGO stages IB2-IVA) [2], 20–40% of these patients will not respond or will develop disease recurrence [3]. The need for further optimization of treatment and improvement of outcome justifies the assessment of tumor-related parameters which could predict response to treatment.

Fluorodeoxyglucose [F18] positron emission tomography combined with computed tomography ([F18]FDG PET/CT) is an important imaging modality in cancer patients [4], [5]. [F18]FDG PET/CT is currently used for the initial diagnosis and evaluation of cervical cancer [2], especially to rule out extrapelvic disease. [F18]FDG PET/CT is more sensitive for the detection of pelvic and distant lymph node metastases in cervical cancer than CT or MRI [6]. There is a direct relationship between the amount of [F18]FDG uptake and the histological type and degree of differentiation of cervical cancer [7]. Moreover, high [F18]FDG uptake in the primary tumor is indicative of higher metastatic potential [8]. Prognostic information related to the extent and metabolic status of the disease in a patient with cervical cancer can be assessed using [F18]FDG PET/CT studies both for the primary tumor and its metastases, whereas assessing response to treatment with PET/CT allows accurate estimation of patient survival [9].

Different tumor metabolic and volumetric parameters can be measured by [F18]FDG PET/CT, both in the primary tumor and its metastases [10]. The most commonly used parameter is the standardized uptake value (SUV), including the SUVmax and SUVmean. Additional parameters include the metabolic tumor volume (MTV) of the disease, and the total lesion glycolysis (TLG), which is obtained by multiplying MTV and SUVmean. MTV and TLG are considered more comprehensive parameters that better reflect the metabolic tumor burden compared with SUV. Tumor volume measurements performed on serial [F18]FDG PET/CT studies after chemo or radiotherapy reflect the shrinkage of the viable tumor and can thus be potentially of higher clinical value compared with anatomic imaging such as CT and MRI [10].

Previous studies have shown that quantitative metabolic and volumetric parameters such as SUVmax, MTV, and TLG in the primary tumor and lymph nodes metastases can help in predicting response to treatment and supply crucial prognostic information in cervical cancer patients, with higher values correlating with poor response to treatment [11,12,13,14,15,16,17,18]. In contrast, a study by Guler et al. found no prognostic significance of metabolic and volumetric parameters such as SUVmax, MTV, and TLG in locally advanced cervical cancer [19].

The current study aimed at exploring the relationship between advanced biological parameters that can be measured using [F18]FDG PET/CT, such as MTV and TLG, and the response to definitive chemoradiation therapy in locally advanced cervical cancer patients. If such a relation exists, additional treatment strategies could be considered in order to increase the cure rate in those patients who are less likely to respond to definitive chemoradiation therapy. To the best of our knowledge, this is the first study aimed at finding cutoff values of [F18]FDG PET/CT-derived parameters that can be used for the prediction of treatment response in locally advanced cervical cancer patients.

## 2. Materials and Methods

Study population: The data of 93 consecutive women with newly diagnosed locally advanced cervical cancer (stages IB2-IVA by the 2009 FIGO classification [20]) between the years 2010–2018, treated with definitive chemoradiation (platinum-based), were retrospectively analyzed. Three patients were excluded from further analysis because of rare tumor histology (such as a neuroendocrine tumor) or faulty PET/CT data storage that did not allow further quantitative measurements. All patients who were included in the study had undergone [F18]FDG PET/CT study for staging and for assessing response to treatment about 3 months after the end of treatment. Patients’ charts were extracted from the institutional database and were reviewed. The following clinical data were recorded: age at the time of diagnosis, tumor histopathology and stage, and data concerning the chemoradiation treatment. This retrospective study was approved by the Ethics Committee of the Rambam HealthCare Campus (permit 0557-19-RMB), and all reported investigations were conducted in accordance with national regulations. Patient informed consent has been waived by the Rambam HealthCare Campus ethic committee because of the retrospective nature of the study.

[F18]FDG PET/CT acquisition and analysis: PET and CT scans (mostly noncontrast enhanced) were acquired consecutively from head to the mid-thigh using a PET/CT system (Discovery 690, GE Healthcare, Milwaukee, WI, USA), at 56–120 min after the injection of average activity 415 MBq (11.2 mCi) [F18]FDG (range: 222 to 617 MBq, 6 to 16.7 mCi).

All PET/CT studies were analyzed by a Nuclear Medicine specialist. Measured parameters included SUVmax and SUVmean in the primary tumor at staging and at 3 months after the completion of treatment, as well as size (short axis) and [F18]FDG uptake parameters (SUVmax and SUVmean) in involved pelvic and paraaortic lymph nodes. Lymph nodes were considered pathological based on their typical location, size, and uptake intensity. In addition, MTV and TLG of all sites of disease were calculated both at staging and post-treatment using the MIRADA XD software (Mirada Medical Ltd., Oxford, UK) with a fixed relative threshold of 41% of SUVmax [10]. In addition to the analysis of the whole study population (FIGO IB2-IVA), a separate analysis was performed in a subgroup of patients with disease limited to the cervix, upper two-thirds of the vagina, and parametrium (FIGO IB2-IIB). For each analysis, patients were divided into two groups—complete metabolic responders (CMR) and noncomplete metabolic responders (n-CMR)—according to the existence of pathological [F18]FDG uptake in the primary tumor or lymph node metastases on PET/CT after treatment completion and also to the final decision of the referring physician.

Statistical analysis: Differences between continuous variables (age, SUVmax, SUVmean, MTV, and TLG) were assessed by the Mann–Whitney test. Differences between categorical parameters (tumor histology) were assessed using the chi-square test. ROC analysis was performed for SUVmax and SUVmean values of the primary tumor and metastatic lymph nodes and for MTV and TLG on the study performed for staging in order to define threshold values that could predict CMR. Statistical analysis was performed in MedCalc software v20.019 by MedCalc Software Ltd., Ostend, Belgium, and *p* value below 0.05 was considered statistically significant.

## 3. Results

Ninety patients with newly diagnosed locally advanced cervical cancer (stages IB2-IVA by FIGO classification 2009) were included in the study. The average age of the patients was 52.8 years (range 29–85). Most patients (88%) had squamous cell carcinoma (SCC) and adenocarcinoma was found in 10 patients (11%).

At staging, 89 patients (99%) showed [F18]FDG uptake in the primary tumor; one patient had no uptake in the primary tumor (of SCC histology) but only in lymph node metastases. The average SUVmax and SUVmean of the primary tumor were 17.1 and 10.6, respectively. Forty-one patients (46%) showed abnormal [F18]FDG uptake in involved lymph nodes, including 32 (78%) in pelvic, 6 (14%) in para-aortic, and 3 (7%) in distant lymph nodes. These patients received definitive chemoradiotherapy to the pelvis and distant lymph nodes. The average size of the lymph nodes was 1.4 cm on the short axis. Average MTV and TLG values were 31.7 cm^3^ and 364.4, respectively.

### 3.1. Total Study Population (90 Patients) Analysis

Fifty-seven patients (63% of the study population) achieved CMR, and 33 patients (37%) were n-CMR. The CMR group included 50 patients (88%) with SCC and 7 (12%) with adenocarcinoma. The n-CMR group included 30 patients (91%) with SCC and 3 (9%) patients with adenocarcinoma. Patients who achieved CMR were significantly younger (*p* < 0.05) than those in the n-CMR group (the average age for the CMR group was 50.2 and for the n-CMR group 57.3). There was no statistically significant difference between the groups in the number of patients with lymph node metastases or in the number of chemotherapy cycles or radiation intensities (Table 1).

Quantitative metabolic and volumetric parameters at staging were extracted. The average SUVmax and SUVmean of the primary tumor were 15.5 and 9.5 in the CMR group and 20.1 and 12.4 in the n-CMR group (*p* < 0.05). Average SUVmax and SUVmean of metastatic lymph nodes were 7.9 and 5.01 in the CMR group and 11.4 and 7.1 in the n-CMR group (*p* < 0.05). Average MTV and TLG were 24.6 cm^3^ and 252.8 for the CMR group and 43.9 cm^3^ and 557.2 for the n-CMR group (*p* < 0.05). A statistically significant difference was found between CMR and n-CMR groups when comparing average SUVmax and SUVmean measurements of the primary tumor and involved lymph nodes as well as MTV and TLG of the entire disease sites together. There was no statistically significant difference in the average size of metastatic lymph nodes between the two groups (Table 2) or their location. The threshold values for SUVmax and SUVmean of the primary tumor were 20.8 (sensitivity 52% and specificity 89%) and 12.1 (61% sensitivity, 79% specificity) and for the involved lymph nodes were 9.2 (68% sensitivity, 74% specificity) and 5.2 (74% sensitivity, 70% specificity). AUC for all of these parameters was calculated as 0.7. Threshold values for MTV and TLG were 31.1 cm^3^ (61% sensitivity, 77% specificity) and 217.8 (79% sensitivity, 63% specificity). AUC values for these parameters were 0.72 and 0.74, respectively.

### 3.2. Analysis of Patients with Disease Limited to the Cervix, Upper Two-Thirds of the Vagina or Parametrium (FIGO Stages IB2-IIB, 48 Patients)

In a subgroup of 48 patients (54%) who had disease limited to the cervix, upper two-thirds of the vagina, or parametrium (FIGO stages IB2-IIB), 38 (79%) had CMR (Figure 1). Ten patients (21%) had residual pathological [F18]FDG uptake in the primary tumor site or had progressed (n-CMR) (Figure 2). In the CMR group, 33 (90%) of patients were diagnosed with SCC, and the rest (5, 10%) were diagnosed with adenocarcinoma. In the n-CMR group, nine (90%) patients were diagnosed with SCC, and one patient (10%) had adenocarcinoma. Demographic data do not differ from those obtained in the entire study population (Table 3). SUVmax for the CMR group was 14.5 and for the n-CMR 18.8, and SUVmean was 9.0 and 11.6, respectively (*p* not significant). MTV values were 18 cm^3^ for the CMR group and 47.2 cm^3^ for the n-CMR group, and TLG values 176 and 582.9, respectively, were significantly different (*p* < 0.05 for both) (Table 4). ROC analysis defined threshold values separating CMR and non-CMR patients with FIGO IB2-IIB disease. The MTV threshold value was 31.1 cm^3^ (70% sensitivity, 91% specificity), and for TLG, it was 217.8 (90% sensitivity and 76% specificity). The area under the curve (AUC) for MTV was 0.87, and for TLG, 0.86 (Figure 3).

## 4. Discussion

Cervical cancer is one of the most common solid malignancies affecting women and a leading cause of cancer-related death in women worldwide. While definitive chemoradiation is the treatment of choice for locally advanced cervical cancer, up to 40% of patients who receive this treatment do not respond as anticipated. Therefore, it is of utmost importance to identify these patients as early as possible and offer them a more appropriate treatment strategy

[F18]FDG PET/CT is an important tool in the evaluation of cervical cancer patients, both for staging and for assessing response to treatment. Defining metabolic and volumetric parameters which could help in predicting response to treatment is of clinical importance. Previous studies [11,12,13,14,15,16,17,18] have emphasized the value of metabolic and volumetric parameters such as SUVmax, MTV, and TLG in the assessment of the primary tumor and of lymph node metastases in providing prognostic information and allowing response prediction. For example, in a study by Hererra et al. [11], which examined 38 cervical cancer patients treated with curative chemoradiation, the authors show that metabolic activity and glycolytic volume of pretreatment [F18]FDG PET/CT can give important prognostic information. In a univariate analysis, mean SUV ≥ 5 and mean TLG ≥ 562 unfavorably influenced overall survival, disease-free survival, and locoregional control. In multivariate analysis, mean TLG ≥ 562 negatively influenced overall survival and disease-free survival. In another study by Lima et al. [18] which examined 82 patients with locally advanced cervical carcinoma treated with definitive chemoradiation, the authors found that pretreatment MTV and TLG and nodal involvement were predictors of response to treatment, with MTV being the best predictor. Higher MTV and TLG values increase the risk of a noncomplete metabolic response to concomitant chemoradiation therapy.

The current study included 90 patients with locally advanced cervical cancer who received definitive chemoradiation therapy and evaluated the role of different metabolic and volumetric parameters of the primary tumor and its metastases in predicting response to treatment. Two analyses were performed, one including the whole study population (FIGO IB2-IVA) and the second only for patients with disease limited to the cervix, upper to two-thirds of the vagina, or parametrium (FIGO IB2-IIB).

The most significant result in the current study was obtained in the subgroup of patients with FIGO IB2-IIB disease at staging. In this group, patients with MTV > 31.1 cm^3^ or TLG > 217.8 on staging PET/CT scan were found to have a high likelihood of residual pathological uptake on [F18]FDG PET. These parameters have an AUC above 0.85, a value that indicates that they are effective in response prediction. However, similar threshold values for prediction of response were not found for SUVmax and SUVmean measurements. It is reasonable to assume that the metabolic volume of the whole tumor has a greater impact on the response to chemoradiation treatment than the metabolic activity measurement in a specific location within the tumor.

When analyzing the whole study population (FIGO IB2-IVA), patients who did not achieve CMR were older than the CMR group. Furthermore, the measured metabolic and volumetric parameters were significantly higher in patients who did not achieve CMR. However, on ROC analysis, most of the parameters achieved values around 0.7, which indicates their relatively lower ability to predict CMR.

The main limitation of this study lies in its retrospective nature. Current results need to be confirmed in a large, prospective study in order to better estimate the role of metabolic and volumetric biomarkers that can be measured by [F18]FDG PET/CT in predicting treatment response in patients with locally advanced cervical cancer. Another limitation is the use of the FIGO 2009 classification since most patients were classified by it initially. This study considered only the complete metabolic response of the primary tumor and lymph node metastasis and not the pathologic response or the overall survival of the patients; this, too, is a limitation of the study.

## 5. Conclusions

Metabolic and volumetric [F18]FDG PET/CT parameters measured at staging in patients with cervical cancer were higher in those who will not completely respond to chemoradiation treatment compared with complete responders. Moreover, if further confirmed in a larger cohort of patients in a subgroup of patients with FIGO IB2-IIB disease, threshold values obtained for MTV and TLG can help in predicting response to treatment and allow adjustment of the therapeutic strategy in those who are expected not to respond.

## Figures and Tables

**Figure 1 cancers-14-04382-f001:**
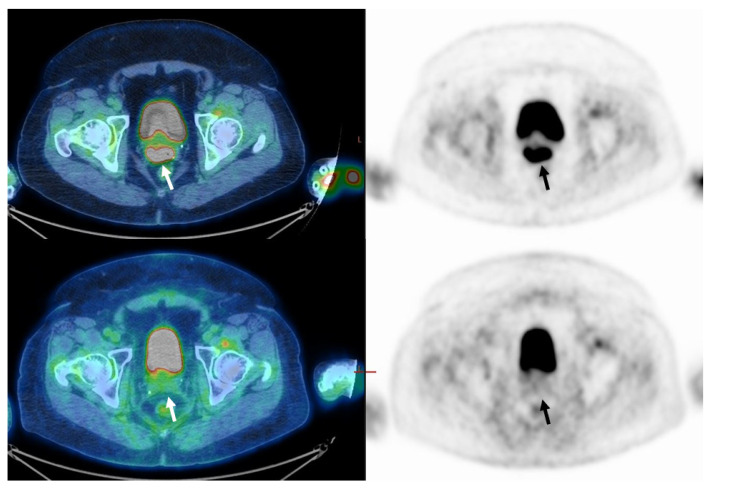
Sixty-two-year-old woman with FIGO stage IIB cervical cancer (Squamous cell carcinoma). On staging, PET/CT images (upper row) demonstrate uptake in the primary tumor only (arrow). MTV was 6.9 cm^3^ and TLG was 65.5. On PET/CT performed about 3 months after the end of treatment (lower row), no pathological uptake is seen (CMR, arrow).

**Figure 2 cancers-14-04382-f002:**
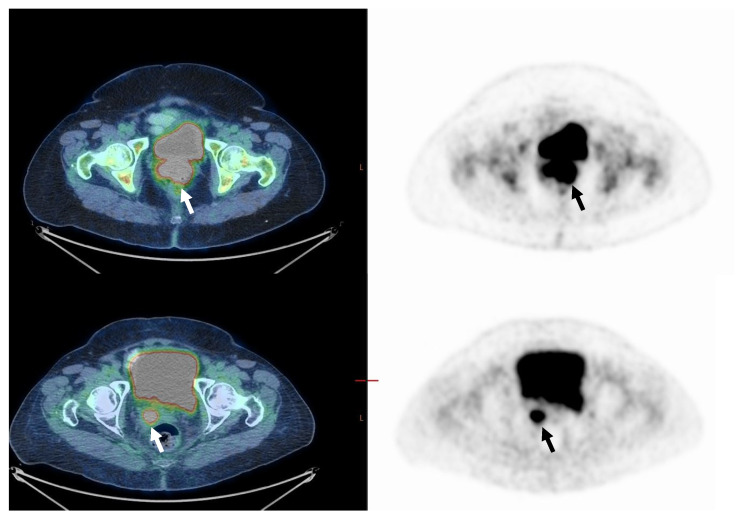
Seventy-year-old woman with FIGO stage IIB disease cervical cancer (Squamous cell carcinoma). On staging, PET/CT images (upper row) demonstrate pathological uptake in the primary tumor only (arrow). MTV was 85.0 cm^3^, and TLG 1411.2. On PET/CT performed about 3 months after the end of treatment (lower row), there is still pathological FDG uptake in the primary tumor (n-CMR, arrow). In addition, a new pathological lymph node has appeared in the right pelvis (not shown in this figure), indicating disease progression.

**Figure 3 cancers-14-04382-f003:**
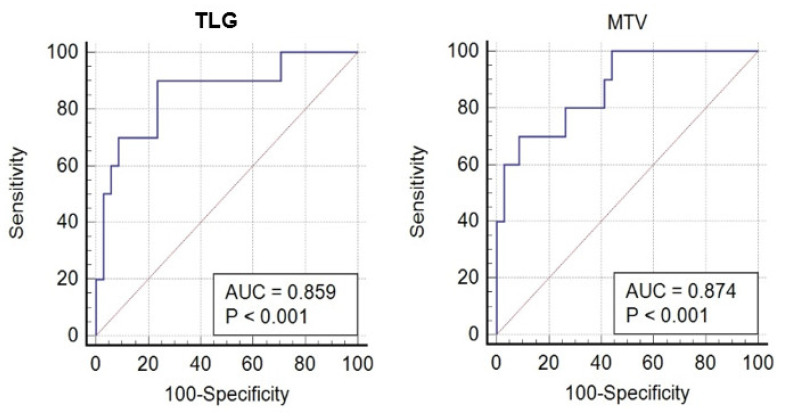
ROC analysis for MTV (**right**) and TLG (**left**) in CMR patients and n-CMR patients FIGO stages IB2-IIB.

**Table 1 cancers-14-04382-t001:** Demographic data of CMR and n-CMR groups. CMR—complete metabolic response, n-CMR—noncomplete metabolic response, N.S.—Nonsignificant; PALN—para-aortic lymph nodes.

Parameter	CMR	n-CMR	*p* Value
Number of patients	57	33	
Average age (Y)	50.2 (29–85)	57.3 (34–82)	*p* < 0.05
The percentage of patients with positive lymph nodes	40.3%	57%	N.S.
Average number of chemotherapy cycles *	4.4	4.1	N.S.
Average external radiation dose (Gray)	49.6	49.7	N.S.
Average brachytherapy dose (Gray)	26.7	25.5	N.S.
Percentage of patients received BOOST/PALN radiation	16%	21%	N.S.

* All patients received platin-based chemotherapy with radiation therapy.

**Table 2 cancers-14-04382-t002:** Metabolic parameters at the staging of patients with CMR and n-CMR. CMR—complete metabolic response, n-CMR—noncomplete metabolic response, SD—standard deviation, SUV—standardized uptake value, MTV—metabolic tumor volume, TLG—total tumor glycolysis.

Parameter	CMR, Average (SD)	n-CMR, Average (SD)	*p* Value
SUVmax of the primary tumor	15.5 (6.3)	20.1 (7.8)	0.003
SUVmean of the primary tumor	9.5 (4.2)	12.4 (5.0)	0.004
Size of lymph nodes * (cm)	1.4 (0.9)	1.4 (0.5)	0.98
SUVmax of metastatic lymph nodes	7.9 (4.8)	11.4 (5.8)	0.03
SUVmean of metastatic lymph nodes	5.01 (2.9)	7.1 (3.6)	0.04
MTV	24.6 (17.7)	43.9 (27.2)	*p* < 0.001
TLG	252.8 (268.8)	557.2 (468.6)	*p* < 0.001

* measured in short axis.

**Table 3 cancers-14-04382-t003:** Demographic and treatment-related data in CMR and n-CMR groups in patients FIGO stages IB2-IIB. CMR—complete metabolic response, n-CMR—noncomplete metabolic response, N.S.—Nonsignificant.

Parameter	CMR	n-CMR	*p* Value
Number of patients	38	10	
Average age (Y)	53 (3–85)	58 (38–71)	N.S.
Average number of chemotherapy cycles *	4.1	4.1	N.S.
Average external radiation dose (Gray)	49.2	49.3	N.S.
Average brachytherapy dose (Gray)	26.5	25.0	N.S.

* All patients received platin-based chemotherapy with radiation therapy.

**Table 4 cancers-14-04382-t004:** Metabolic parameters in CMR and n-CMR groups in patients FIGO stages IB2-IIB. CMR—complete metabolic response, n-CMR—noncomplete metabolic response, SD—standard deviation, SUV—standardized uptake value, MTV—metabolic tumor volume, TLG—total tumor glycolysis.

Parameter	CMR, Average (SD)	n-CMR, Average (SD)	*p* Value
SUVmax of the primary tumor	14.5 (6.3)	18.8 (7.5)	0.06
SUVmean of the primary tumor	9.0 (4.5)	11.6 (4.8)	0.07
MTV	18 (11.7)	47.2 (25.0)	*p* < 0.001
TLG	176 (181.7)	582.9 (434.1)	*p* < 0.001

## Data Availability

The data presented in this study are available in this article.

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
