# Peer review of "[F18]FDG PET/CT-Derived Metabolic and Volumetric Biomarkers Can Predict Response to Treatment in Locally Advanced Cervical Cancer Patients"

_cancers, 2022, doi:10.3390/cancers14184382_

Round 1

Reviewer 1 Report

I like to congratulate the authors for the nice manuscript. I have few suggestion;

1. even through it is mentioned at the end of manuscript ,that they have used FIGO 2009 staging , It will be better to mention this at the beginning and perhaps define their tumor size at the time of staging.

2. three patients had distant metastasis, please define if they also had complete local chemoradiation with cure intent or it was palliative radiation . Did they receive further systemic therapy as well?

3. Why did authors choose the subcategory of Stage IB2 -IIB. Please state the reasoning and speculate or explain why stage IVA did not show same results.

4. Line 267,  It means to be  with   and  not  wuth. 

Author Response

Response to Reviewer 1 Comments

Point 1: Even though it is mentioned at the end of manuscript, that they have used FIGO 2009 staging , It will be better to mention this at the beginning and perhaps define their tumor size at the time of staging. 

Response 1: Thank you for this comment. The FIGO classification from 2009 is mentioned in "Materials and Methods", p.2, line 87- " stages IB2-IVA by the 2009 FIGO classification". Tumor size measurement was avoided because of different CT techniques (with and without IV contrast enhancement).

Point 2:  Three patients had distant metastasis, please define if they also had complete local chemoradiation with cure intent or it was palliative radiation. Did they receive further systemic therapy as well?

Response 2: Patients had distant LN metastases (paraaortic, inguinal, left supraclavicular), received definitive chemo-radiotherapy to pelvis, distant lymph nodes. The following sentence was added to the Results section “These patients received definitive chemo-radiotherapy to pelvis and distant lymph nodes”

Point 3: Why did authors choose the subcategory of Stage IB2-IIB. Please state the reasoning and speculate or explain why stage IVA did not show same results

Response 3: We thought that this subgroup of patients with only local disease without any metastases will be of interest.

Point 4: Line 267,  It means to be  with   and  not  wuth

Response 4: Thank you. Was corrected.

Reviewer 2 Report

The current study evaluates the relationship between [F18]FDG PET/CT metabolic and volumetric parameters such as MTV and TLG and the response to treatment in 90 patients with (FIGO IB2-IVA) locally advanced cervical cancer who received definitive chemo-radiation therapy . This parameters measured at staging were higher in patients non-CMR as compared to complete responders.  A second analysis was performed for a subgroup of 48 patients with limited disease  (FIGO IB2-IIB), finding cut off values of MTV and TLG that, if further confirmed in a larger cohort, can help in predicting the cure rate.

The article is well-organized and contains all the required components, it's well-written and easy to understand. The authors answer the research questions in a satisfying manner and the theory is connect to the data. The literature is consistent whit the authors job, even if it's a bit dated. Some references could be added in the Discussion, especially about threshold calculation, the use of AUC, ROC and their meaning and relationship whit the specific desease.

The methodology is clearly explained and the sections are developed accurately, but Discussion could be more detailed in some points:

Why, as opposed to the other parameters, in Table 2 (all patients) TLGmean is greater than TLGthreshold, ?

Clarify authors' statement on line 281-283 "It is reasonable to assume that the metabolic volume of the tumor has a greater impact on the response to chemo-radiation treatment than the metabolic activity in a certain point in the tumor or average metabolic activity of the entire tumor".

Explain why the autors claim that the use of the FIGO 2009 classification it's a limitation for the study, and what type of classification could be better.

The research design is good, with homogeneous data set regarding treatments, number and size of metastatic lymph nodes, type of disease and demographic data. Tables and figures are clear and complete.

As reported by the authors, to strengthen the work, results need to be confirmed in a large, prospective study whit additional investigations on overall survival.

The authors wrote “sxis” on line 108 and "wuth" on line 267.

Author Response

Point 1: Why, as opposed to the other parameters, in Table 2 (all patients) TLG mean is greater than TLG threshold ?

Response 1: Thank you for this comment. This discrepancy is probably related to the “fair” AUC value of this parameter (0.74).

Point 2: Clarify authors' statement on line 281-283 "It is reasonable to assume that the metabolic volume of the tumor has a greater impact on the response to chemo-radiation treatment than the metabolic activity in a certain point in the tumor or average metabolic activity of the entire tumor".

Response 2: The sentence was rephrased to be clearer and it is now “It is reasonable to assume that the metabolic volume of the whole tumor has a greater impact on the response to chemo-radiation treatment than the metabolic activity measurement within a specific location in the tumor.”

Point 3: Explain why the autors claim that the use of the FIGO 2009 classification it's a limitation for the study, and what type of classification could be better.

Response 3:  FIGO 2009 classification did not included LN involvement into the staging. Newer staging classify pelvic LN involvement as stage IIIC1, paraaortic LN involvement as stage IIIC2 disease. 

Point 4: The authors wrote “sxis” on line 108 and "wuth" on line 267.

Response 4:  Thank you for this comment. It was corrected.